# Learning Deep Input-Output Stable Dynamics

**Ryosuke Kojima**[*]
Graduate School of Medicine
Kyoto University
Kyoto, 606-8501
kojima.ryosuke.8e@kyoto-u.ac.jp

**Yuji Okamoto**[*]
Graduate School of Medicine
Kyoto University
Kyoto, 606-8501
okamoto.yuji.2c@kyoto-u.ac.jp

## Abstract

Learning stable dynamics from observed time-series data is an essential problem in robotics, physical modeling, and systems biology. Many of these dynamics are represented as an inputs-output system to communicate with the external environment. In this study, we focus on input-output stable systems, exhibiting robustness against unexpected stimuli and noise. We propose a method to learn nonlinear systems guaranteeing the input-output stability. Our proposed method utilizes the differentiable projection onto the space satisfying the Hamilton-Jacobi inequality to realize the input-output stability. The problem of finding this projection can be formulated as a quadratic constraint quadratic programming problem, and we derive the particular solution analytically. Also, we apply our method to a toy bistable model and the task of training a benchmark generated from a glucose-insulin simulator. The results show that the nonlinear system with neural networks by our method achieves the input-output stability, unlike naive neural networks. Our code is available at https://github.com/clinfo/DeepIOStability.

## 1 Introduction

Learning dynamics from time-series data has many applications such as industrial robot systems [1], physical systems[2], and biological systems [3, 4]. Many of these real-world systems equipped with inputs and outputs to connect for each other, which are called *input-output systems* [5]. For example, biological systems sustain life by obtaining energy from the external environment through their inputs. Such real-world systems have various properties such as stability, controllability, and observability, which provide clues to analyze the complex systems.

Our purpose is to learn a complex system with "desired properties" from a dataset consisting of pairs of input and output signals. To represent the target system, this paper considers the following nonlinear dynamics:

$$\begin{aligned} \dot{x} &= f(x) + G(x)u, \quad x(0) = x_0 \\ y &= h(x). \end{aligned} \tag{1}$$

where the inner state $x$, the input $u$, and the output $y$ belong to a signal space that maps time interval $[0, \infty)$ to the Euclidean space. We denote the dimension of $x$, $u$, and $y$ as $n$, $m$, and $l$, respectively.

Recently, with the development of deep learning, many methods to learn systems from time-series data using neural networks have been proposed [6–8]. By representing the maps $(f, G, h)$ in Eq. (1) as neural networks, complex systems can be modeled and trained from a given dataset. However, guaranteeing that a trained system has the desired properties is challenging.

A naively trained system fits the input signals contained in the training dataset, but does not always fit for new input signals. For example, Figure 1 shows our preliminary experiments where we

---

[*]Equal contribution.

36th Conference on Neural Information Processing Systems (NeurIPS 2022).

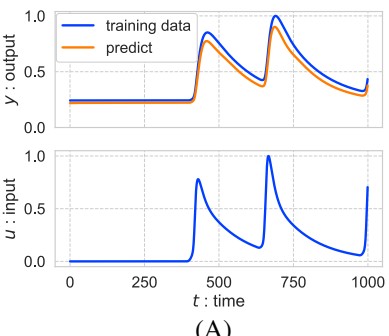
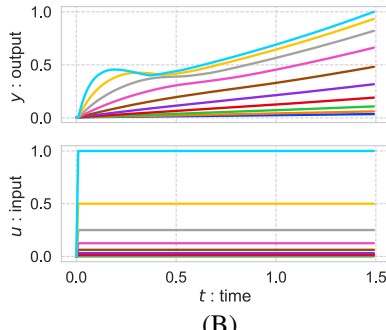

(A)              (B)

Figure 1: (A) The model prediction of neural networks (B) the reaction of trained model. These results are min-max normalized.

naively learned neural networks $(f, G, h)$ in Eq. (1) from input and output signals. The trained neural networks provided small predictive errors for an input signal in the training dataset (Figure 1 (A) ). Contrastingly, the unbounded output signals were computed by the trained system with new step input signals (Figure 1 (B) ). The reason can be expected that the magnitude (integral value) of this input signal was larger than that of the input signals in the training dataset.

The internal stability is known as one of the attractive properties that should be often satisfied in real-world dynamical systems. The conventional methods to train internal stable systems consisting of neural networks adopt the Lyapunov-based approaches [9]–[11]. These methods focus on the internal system, $\dot{x} = f(x)$ in the input-output system (1). Thus, how to learn the entire system (1) with the desired property related to the influence of the input signal is still challenging.

We propose a novel method to learn a dynamical system consisting of neural networks considering the input-output stability. The notion of the input-output stability is often used together with the Hamilton-Jacobi inequality for controller design of the target system in the field of control theory. The Hamilton-Jacobi inequality is one of the sufficient conditions for the input-output stability. The feature of this condition is that the variable for an input signal $u$ does not appear in the expression, i.e., we do not need to evaluate the condition for unknown inputs [12]. To the best of our knowledge, this is the first work that establishes a learning method for the dynamical systems consisting of neural networks using the Hamilton-Jacobi inequality.

The contributions of this paper are as follows:

- This paper derives differentiable projections to the space satisfying the Hamilton-Jacobi inequality.

- This paper presents the learning method for the input-output system proven to always satisfy the Hamilton-Jacobi inequality.

- This paper also provides a loss function derived from the Hamilton-Jacobi inequality. By combining this loss function with the projection described above, efficient learning can be expected.

- This paper presents experiments using two types of benchmarks to evaluate our method.

## 2 Background

This section describes the $\mathcal{L}_2$ stability, a standard definition of the input-output stability, and the Hamilton-Jacobi inequality.

First, we define the $\mathcal{L}_2$ stability of the nonlinear dynamical system (1). If the norm ratio of the output signal to the input signal is bounded, the system is $\mathcal{L}_2$ stable and this norm ratio is called the $\mathcal{L}_2$ gain. The $\mathcal{L}_2$ norm on the input and output signal space is used for the definition of the $\mathcal{L}_2$ stability. To deal with the $\mathcal{L}_2$ stability on the nonlinear system (1), we assume that the origin $x \equiv 0$ of the internal system $\dot{x} = f(x)$ is an asymptotically stable equilibrium point and $h(0) = 0$. Since the translation of

any asymptotically stable equilibrium points is possible, this assumption can be satisfied without loss of generality.

**Definition 1.** *The $\mathcal{L}_2$ norm is defined as $\|x\|_{\mathcal{L}_2} := \sqrt{\int_0^\infty \|x(t)\|^2 \mathrm{d}t}$. If there exists $\gamma \geq 0$ and a function $\beta$ on a domain $D \subset \mathbb{R}^n$ such that*

$$\|y\|_{\mathcal{L}_2} \leq \gamma \|u\|_{\mathcal{L}_2} + \beta(x_0), \tag{2}$$

*then the system (1) is $\mathcal{L}_2$ stable, where the function $\beta(\cdot)$ is non-negative and $\beta(0) = 0$. Furthermore, the minimum $\gamma$ satisfying (2) is called the $\mathcal{L}_2$ gain of the nonlinear system (1).*

Next, we describe a sufficient condition of the $\mathcal{L}_2$ stability using the Lyapunov function $V : D \to \mathbb{R}$, where the function $V$ is positive semi-definite, i.e., $V(x) \geq 0$ and $V(0) = 0$. The Hamilton-Jacobi inequality is an input-independent sufficient condition.

**Proposition 1** ([5, Theorem 5.5]). *Let $f$ be locally Lipschitz and let $G$ and $h$ be continuous. If there exist a constant $\gamma > 0$ and a continuously differentiable positive semi-definite function $V : D \subset \mathbb{R}^n \to \mathbb{R}$ such that*

$$\nabla V^{\mathrm{T}}(x)f(x) + \frac{1}{2\gamma^2}\|G^{\mathrm{T}}(x)\nabla V(x)\|^2 + \frac{1}{2}\|h(x)\|^2 \leq 0, \quad \forall x \in D \setminus \{0\}, \tag{3}$$

*then the system (1) is $\mathcal{L}_2$ stable and the $\mathcal{L}_2$ gain is less than or equal to $\gamma$. This condition is called the Hamilton-Jacobi inequality.*

The above proposition can be generalized to allow the more complicated situations such as limit-cycle and bistable cases, where the domain $D$ contains multiple asymptotically stable equilibrium points. The equilibrium point assumed in this proposition can be replaced with positive invariant sets by extending the $\mathcal{L}_2$ norm [13]. Furthermore, by mixing multiple Lyapunov functions, this proposition can be generalized around multiple isolated equilibrium points.

## 3 Method

The goal of this paper is to learn the $\mathcal{L}_2$ stable system represented by using neural networks $(f, G, h)$. The Hamiltonian-Jacobi inequality, which implies the $\mathcal{L}_2$ stability, is expressed by $(f, G, h)$. We present a method to project $(f, G, h)$ onto the space where the Hamilton-Jacobi inequality holds.

### 3.1 Modification of nonlinear systems

Supposing $f_{\mathbf{n}} : \mathbb{R}^n \to \mathbb{R}^n$, $G_{\mathbf{n}} : \mathbb{R}^n \to \mathbb{R}^{n \times m}$, and $h_{\mathbf{n}} : \mathbb{R}^n \to \mathbb{R}^l$, a triplet map $(f_{\mathbf{n}}, G_{\mathbf{n}}, h_{\mathbf{n}})$ denote as *nominal* dynamics. Introducing a distance on the triplet maps, the nearest triplet satisfying the Hamilton-Jacobi inequality from the nominal dynamics $(f_{\mathbf{n}}, G_{\mathbf{n}}, h_{\mathbf{n}})$ is called modified dynamics $(f_{\mathbf{m}}, G_{\mathbf{m}}, h_{\mathbf{m}})$. The purpose of this section is to describe the modified dynamics $(f_{\mathbf{m}}, G_{\mathbf{m}}, h_{\mathbf{m}})$ associated with the nominal dynamics $(f_{\mathbf{n}}, G_{\mathbf{n}}, h_{\mathbf{n}})$ by analytically deriving a projection onto the space satisfying the Hamilton-Jacobi inequality.

The problem of finding the modified dynamics $(f_{\mathbf{m}}, G_{\mathbf{m}}, h_{\mathbf{m}})$ is written as a quadratic constraint quadratic programming (QCQP) problem for the nominal dynamics $(f_{\mathbf{n}}, G_{\mathbf{n}}, h_{\mathbf{n}})$. Since there is generally no analytical solution for QCQP problems, we aim to find the particular solution by adjusting the distance on the triplets.

To prepare for the following theorem, we define the ramp and the clamp functions as

$$\mathrm{R}(x) \triangleq \begin{cases} 0, & x \leq 0 \\ x, & x > 0 \end{cases}, \quad \mathrm{C}(x; a, b) \triangleq \begin{cases} a, & x \leq a \\ x, & a < x \leq b \\ b, & x > b \end{cases},$$

and define the Hamilton-Jacobi function as

$$\mathrm{HJ}(f, G, h) \triangleq \nabla V^{\mathrm{T}}f + \frac{1}{2\gamma^2}\|G^{\mathrm{T}}\nabla V\|^2 + \frac{1}{2}\|h\|^2, \tag{4}$$

where $V$ is a given positive definite function. A way to design $V$ is to determine the desired positive invariant set and design the increasing function around this set. For example, if the target system has a unique stable point, $x = 0$ can be regarded as the positive invariant set and the increasing function $V$ can be designed as $V(x) = \frac{1}{2}x^2$.

**Theorem 1.** *Consider that the following optimal problem:*

$$\underset{f_{\mathbf{m}}, G_{\mathbf{m}}, h_{\mathbf{m}}}{\textbf{minimize}} \quad \frac{(1-k)}{\|\nabla V\|}\|f_{\mathbf{m}} - f_{\mathbf{n}}\| + \frac{k}{2\gamma^2}\|G_{\mathbf{m}} - G_{\mathbf{n}}\|^2 + \frac{k}{\|\nabla V\|^2}\|h_{\mathbf{m}} - h_{\mathbf{n}}\|^2 \quad (5a)$$

$$\textbf{subject to} \quad \text{HJ}(f_{\mathbf{m}}, G_{\mathbf{m}}, h_{\mathbf{m}}) \leq 0, \quad (5b)$$

*where $k \in [0,1]$. The solution of (5) is given by*

$$f_{\mathbf{m}} = f_{\mathbf{n}} - \frac{1}{\|\nabla V\|^2}\text{R}\left(V_f + k^2 V_{Gh}\right)\nabla V,$$

$$G_{\mathbf{m}} = G_{\mathbf{n}} - \left(1 - \sqrt{\text{C}\left(-\tfrac{V_f}{V_{Gh}}; k^2, 1\right)}\right) P_V G_{\mathbf{n}},$$

$$h_{\mathbf{m}} = \sqrt{\text{C}\left(-\tfrac{V_f}{V_{Gh}}; k^2, 1\right)} h_{\mathbf{n}},$$

*where*

$$V_f \triangleq \nabla V^{\text{T}} f_{\mathbf{n}}, \quad V_{Gh} \triangleq \frac{1}{2\gamma^2}\|G_{\mathbf{n}}^{\text{T}}\nabla V\|^2 + \frac{1}{2}\|h_{\mathbf{n}}\|^2, \quad P_V \triangleq \frac{\nabla V \nabla V^{\text{T}}}{\|\nabla V\|^2}.$$

**Proof:** See Appendix A. □

The objective function of (5a) is a new distance between the nominal dynamics $(f_{\mathbf{n}}, G_{\mathbf{n}}, h_{\mathbf{n}})$ and the modified dynamics $(f_{\mathbf{m}}, G_{\mathbf{m}}, h_{\mathbf{m}})$. This new distance allows the derivation of analytical solutions by combining three distances of $f$, $G$, and $h$.

Focusing on the objective function (5a), the constant $k$ represents the ratio of the distance scale of $f$ to $G$ and $h$. When $k = 0$, the result of this problem (5) are consistent with the projection of the conventional method that guarantee internal stability [9]. As the constant $k$ converges 1, the modification method of Theorem 1 approaches $G_{\mathbf{m}}$ and $h_{\mathbf{m}}$ to $G_{\mathbf{n}}$ and $h_{\mathbf{n}}$, respectively. In this case, the objective function (5a) becomes the distance between $f_{\mathbf{m}}$ to $f_{\mathbf{n}}$. Therefore, the following corollary is satisfied.

**Corollary 1.** *The solution of*

$$\underset{f_{\mathbf{m}}}{\textbf{minimize}} \quad \|f_{\mathbf{m}} - f_{\mathbf{n}}\| \quad (6a)$$

$$\textbf{subject to} \quad \text{HJ}(f_{\mathbf{m}}, G_{\mathbf{n}}, h_{\mathbf{n}}) \leq 0, \quad (6b)$$

*is given by*

$$f_{\mathbf{m}} = f_{\mathbf{n}} - \frac{1}{\|\nabla V\|^2}\text{R}\left(\text{HJ}(f_{\mathbf{n}}, G_{\mathbf{n}}, h_{\mathbf{n}})\right)\nabla V.$$

**Proof:** This solution is easily derived from Theorem 1. □

Corollary 1 derives a solution of a linear programming problem rather than QCQP problems. Replacing the Hamilton-Jacobi function $\text{HJ}(f_{\mathbf{m}}, G_{\mathbf{n}}, h_{\mathbf{n}})$ with the time derivative of a positive definite function $\nabla V^{\text{T}} f$, this corollary matches the result of the conventional study [9].

When the map $h_{\mathbf{m}}$ is fixed as $h_{\mathbf{n}}$, a similar solution as Theorem 1 is derived. Although Corollary 1 is proved by changing $k$, the modified dynamics with the fixed $h_{\mathbf{n}}$ are not derived. We reprove this modified dynamics in a similar way to Theorem 1.

**Corollary 2.** *Consider the following problem:*

$$\underset{f_{\mathbf{m}}, G_{\mathbf{m}}}{\textbf{minimize}} \quad \frac{(1-k)}{\|\nabla V\|}\|f_{\mathbf{m}} - f_{\mathbf{n}}\| + \frac{k}{2\gamma^2}\|G_{\mathbf{m}} - G_{\mathbf{n}}\|^2 \quad (7a)$$

$$\textbf{subject to} \quad \text{HJ}(f_{\mathbf{m}}, G_{\mathbf{m}}, h_{\mathbf{n}}) \leq 0, \quad (7b)$$

*where $k \in [0,1]$. The solution of (7) is given by*

$$f_{\mathbf{m}} = f_{\mathbf{n}} - \frac{1}{\|\beta\|^2}\text{R}\left(V_{fh} + k^2 V_G\right)\nabla V, \quad G_{\mathbf{m}} = G_{\mathbf{n}} - \left(1 - \sqrt{\text{C}\left(-\tfrac{V_{fh}}{V_G}; k^2, 1\right)}\right) P_V G_{\mathbf{n}}, \quad (8)$$

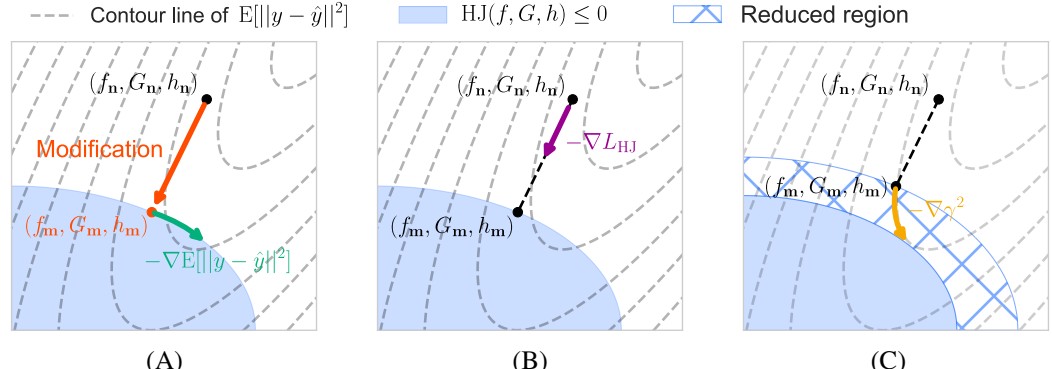

Figure 2: Sketches of our method : (A) minimizing the prediction error (the first term of the loss function (9)) in the blue region, (B) moving the nominal dynamics to the blue region (the second term), and (C) reducing the blue region while keeping the same level of the prediction error (the last term).

*where*

$$V_{fh} \triangleq \nabla V^{\mathrm{T}} f_{\mathbf{n}} + \frac{1}{2}\|h_{\mathbf{n}}\|^2, \quad V_G \triangleq \frac{1}{2\gamma^2}\|G_{\mathbf{n}}^{\mathrm{T}}\nabla V\|^2.$$

**Proof:** See Appendix A. $\qquad\square$

### 3.2 Loss function

We represent $(f_{\mathbf{n}}, G_{\mathbf{n}}, h_{\mathbf{n}})$ as neural networks and denote $(f_{\mathbf{m}}, G_{\mathbf{m}}, h_{\mathbf{m}})$ as the $\mathcal{L}_2$ stable dynamics modified by Theorem 1, Corollary 1, or 2. Note that the modification depends on the candidate of $\mathcal{L}_2$ gain $\gamma$. Figure 2 (A) shows the sketch of this modification, where the blue region satisfies the Hamilton-Jacobi inequality.

Since the nonlinear system of the modified dynamics $(f_{\mathbf{m}}, G_{\mathbf{m}}, h_{\mathbf{m}})$ is represented as ordinary differential equations (ODEs) consisting of the differentiable functions $f_{\mathbf{n}}, G_{\mathbf{n}}$, and $h_{\mathbf{n}}$, the techniques of training neural ODEs can be applied [14]. Once a loss function is designed, the parameters of neural networks in the modified system can be learned from given data.

$$\text{Loss} = \mathrm{E}_{(x_0, u, \hat{y}) \in \mathcal{D}}[\|y - \hat{y}\|_{\mathcal{L}_2}^2] + \lambda L_{\mathrm{HJ}} + \alpha\gamma^2, \tag{9}$$

where $\lambda$ and $\alpha$ are non-negative coefficients.

The first term shows the prediction error of the output signal $y$ ( Figure 2 (A)). A dataset $\mathcal{D}$ consists of tuples $(x_0, u, y)$ where the initial value $x_0$, the input signal $u$, and the output signal $y$. The predicted output $\hat{y}$ is calculated from $x_0$, $u$, and the modified dynamics $(f_{\mathbf{m}}, G_{\mathbf{m}}, h_{\mathbf{m}})$.

The second term aims to improve the nominal dynamics $(f_{\mathbf{n}}, G_{\mathbf{n}}, h_{\mathbf{n}})$ closer to the modified dynamics $(f_{\mathbf{m}}, G_{\mathbf{m}}, h_{\mathbf{m}})$ and is defined as

$$L_{\mathrm{HJ}} = \mathrm{E}_x[\mathrm{R}(\mathrm{HJ}(f_{\mathbf{n}}, G_{\mathbf{n}}, h_{\mathbf{n}})(x) + \varepsilon)],$$

where $\varepsilon$ is a positive constant ( Figure 2 (B)). Since this term is a form of the hinge loss, $\varepsilon$ represents the magnitude of the penalties for the Hamilton-Jacobi inequality. To evaluate this inequality for any $x \in D$, we introduce a distribution of $x$ over the domain $D$. In our experiments, this distribution is decided as a Gaussian distribution $\mathcal{N}(\mu, \sigma^2)$ where the mean $\mu$ is placed at the asymptotically stable point and the variance $\sigma^2$ is an experimental parameter. Without this second term of the loss function, there are degrees of freedom in the nominal dynamics, i.e., multiple nominal dynamics give the same loss by the projection, which negatively affects the parameter learning.

The modifying parameter $\gamma$ can be manually designed for the application or automatically trained from data by introducing the last term. This training explores smaller $\gamma$ while keeping the same level of the prediction error (Figure 2 (C)).

# 4 Related work

Estimating parameters of a given system is traditionally studied as system identification. In the field of system identification, much research on the identification of linear systems have been done, where the maps $(f, G, h)$ in Eq. (1) assumes to be linear [15]. Linear state-space models include identification methods by impulse response like the Eigensystem Realization Algorithm (ERA)[16], by the state-space representation like Multivariable Output Error State sPace (MOESP)[17] and ORThogonal decomposition method (ORT)[18]. System identification methods for non-state-space models, unlike our target system, contain Dynamic Mode Decomposition with control (DMDc)[19] and its nonlinear version, Sparse Identification of Nonlinear DYnamics with control (SINDYc)[20]. Also, the nonlinear models such as the nonlinear ARX model [21] and the Hammerstein-Wiener model [22] have been developed and often trained by error minimization. For example, the system identification method using a piece-wise ARX model allows more complicated functions by relaxing the assumption of linearity [21].

These traditional methods often concern gray-box systems, where the maps $(f, G, h)$ in Eq. (1) are partially known [23]. This paper deals with a case of black-box systems, where $f$, $G$, and $h$ in the system (1) are represented by neural networks. Our method can be used regardless of whether all $f$, $G$, and $h$ functions are parameterized using neural networks. So, our method can be easily applied to application-dependent gray-box systems when the functions $f$, $G$, and $h$ are differentiable with respect to the parameters.

Because the input-output system (1) can be regarded as a differential equation, our study is closely related to the method of combining neural networks and ODEs [7, 14]. These techniques have been improved in recent years, including discretization errors and computational complexity. Although we used an Euler method for simplicity, we can expect that learning efficiency would be further improved by using these techniques.

The internal stability is a fundamental property in ODEs, and learning a system with this property using neural networks plays an important role in the identification of real-world systems [24]. In particular, the first method to guarantee the internal stability of the trained system has been proposed in [9]. Furthermore, another method [25] extends this method to apply positive invariant sets, e.g. limit cycles and line attractors. Encouraged by these methods based on the Lyapunov function, our method further generalizes these methods using the Hamilton-Jacobi inequality to guarantee the input-output stability.

In this paper, the Lyapunov function $V$ is considered to be given, but this function can be learned from data. Since a pair of dynamics (1) and $V$ has redundant degrees of freedom, additional assumptions are required to determine $V$ uniquely. In [9], it is realized by limiting the dynamics to the internal system and restricting $V$ to a convex function [26].

Lyapunov functions are also used to design controllers, where the whole system should satisfy the stability condition. A method for learning such a controller using neural nets has been proposed [27]. This method deals with optimization problems over the space that satisfies the stability condition, which is similar to our method. Whereas we solve the QCQP problem to derive the projection onto the space, this method uses a Satisfiability Modulo Theories (SMT) solver to satisfy this condition. The method has also been extended to apply unknown systems [28].

Although this paper deals with deterministic systems, neural networks for stochastic dynamics with the variational inference is well studied [6, 29–31]. Guaranteeing the internal stability of trained stochastic dynamics is important for noise filtering and robust controller design [10].

# 5 Experiments

We conduct two experiments to evaluate our proposed method. The first experiment uses a benchmark dataset generated from a nonlinear model with multiple asymptotically equilibrium points. In the next experiment, we applied our method to a biological system using a simulator of the glucose-insulin system.

## 5.1 Experimental setting

Before describing the results of the experiments, this section explains the evaluation metrics and our experimental setting.

For evaluation metrics, we define the root mean square error (RMSE) and the average $\mathcal{L}_2$ gain (GainIO) for the given input and output signals as follows:

$$\text{RMSE} \triangleq \sqrt{\frac{1}{N} \sum_{i=1}^{N} \|y_i - \hat{y}_i\|_{\mathcal{L}_2}^2}, \quad \text{GainIO} \triangleq \frac{1}{N} \sum_{i=1}^{N} \frac{\|\hat{y}_i\|_{\mathcal{L}_2}}{\|u_i\|_{\mathcal{L}_2}},$$

where $N$ is the number of signals in the dataset. $u_i(\cdot)$ and $y_i(\cdot)$ are the input and output signal at the $i$-th index, respectively. The prediction signal $\hat{y}_i(\cdot)$ is computed from $u_i(\cdot)$, the trained dynamics, and the initial state. Note that the integral contained in the $\mathcal{L}_2$ norm is approximated by a finite summation. The RMSE is a metric of the prediction errors related to the output signal, and the GainIO is a metric of the property of the $\mathcal{L}_2$ stability. Whether the target system satisfies or does not satisfy the $\mathcal{L}_2$ stability, the GainIO with a given finite-size dataset can be calculated. The GainIO error is defined by the absolute error between the GainIO of the test dataset and that of the prediction.

In our experiments, 90% of the dataset is used for training and the remaining 10% is used for testing. We retry five times for all experiments and show the mean and standard deviations of the metrics.

For simplicity in our experiments, the sampling step $\Delta t$ for the output $y$ is set as constant and the Euler method is used to solve ODEs. $x_0$ is put at an asymptotically stable point for each benchmark and is known. In this experiment, to prevent the state from diverging during learning of dynamics, the clipping operation is used so that the absolute values of the states are less than ten.

For comparative methods, we use vanilla neural networks, ARX, ORT [18], MOESP [17], and piece-wise ARX[21]. In the method of vanilla neural networks, the maps $(f, G, h)$ in the nonlinear system (1) is represented by using three neural networks, i.e., this method is consistent with a method used in Figure 1. To determine the hyperparameters of comparative methods except for neural networks, the grid search is used. Note that these comparative methods only consider the prediction errors.

For training each method with neural networks, an NVIDIA Tesla T4 GPU was used. Our experiments are totally run on 20 GPUs over about three days.

## 5.2 Bistable model benchmark

The first experiment is carried out using a bistable model, which is known as a bounded system with multiple asymptotically equilibrium points. This bistable model is defined as

$$\dot{x} = x(1 - x^2) + u, \quad x(0) = -1, \quad y = x.$$

The internal system of this model has two asymptotically stable equilibrium points $x \equiv 1, -1$.

We generate 1000 input and output signals for this experiment. To construct this dataset, we prepare input signals using positive and negative pulse wave signals whose pulse width is changed at random. The input and output signals on the period $[0, 10]$ are sampled with an interval $\Delta t = 0.1$. In this benchmark, we set the number of dimensions of the internal system as one and use a fixed function $V(x) = \min((x - 1)^2, (x + 1)^2)$, a mixture of the two positive definite functions.

In the result of our experiments, we name the proposed methods modified by Theorem 1, Corollary 1, and 2 as DIOS-fgh, DIOS-f, and DIOS-fg, respectively. In these methods, the parameters of our loss function are set as $\lambda = 0$ and $\alpha = 0.01$. Also, DIOS-fgh+ uses $\lambda = 0.01$ and $\alpha = 0.01$ under the same conditions as DIOS-fgh. For this example (1000 input signal), it took about 1 hour using 1 GPU to learn one model training.

The results of the RMSE and the GainIO error in this experiment are shown in Figure 3. Figures 3 (A) and (B) demonstrate that a piece-wise ARX model (PWARX) gives very low RMSE but its GainIO error is high. Our proposed methods achieve a small GainIO error while keeping the RMSE. Note that linear models such as MOESP and ORT only approach one point although the bistable model has two asymptotically stable equilibrium points. Figures 3 (C) and (D) display the effect of dataset sizes. When the dataset size was varied to 100, 1000, and 3000, the result of the larger dataset provided

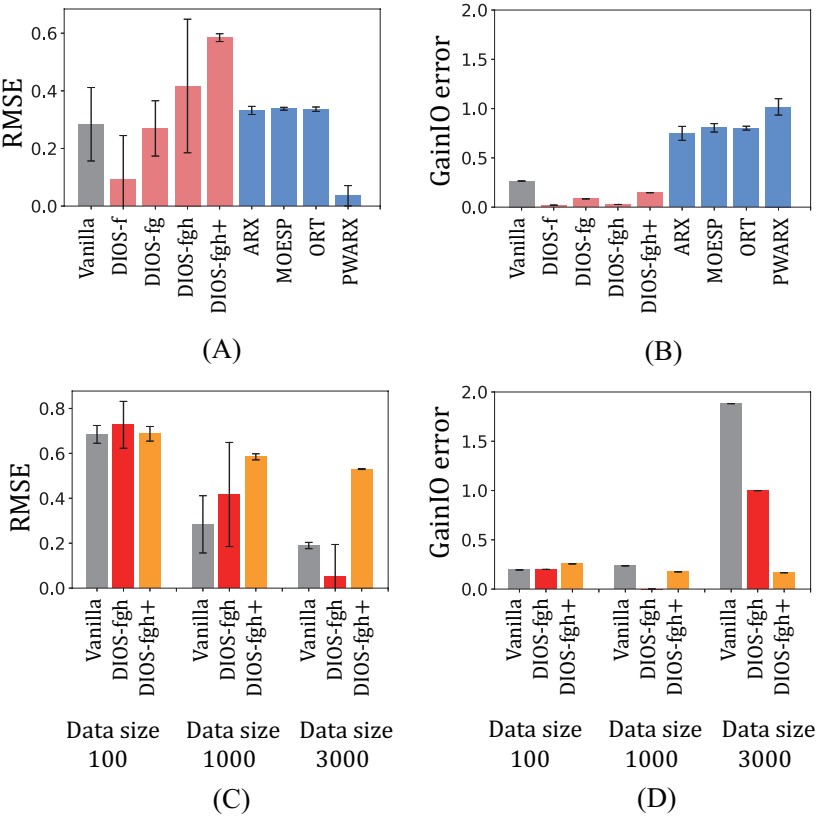

Figure 3: Results of the bistable model benchmark. The upper part shows (A) the RMSE and (B) the GainIO error of the vanilla neural networks (gray), our proposed methods (red), and the conventional methods (blue). The lower part shows (C) the RMSE and (D) the GainIO error for the different sizes of the datasets.

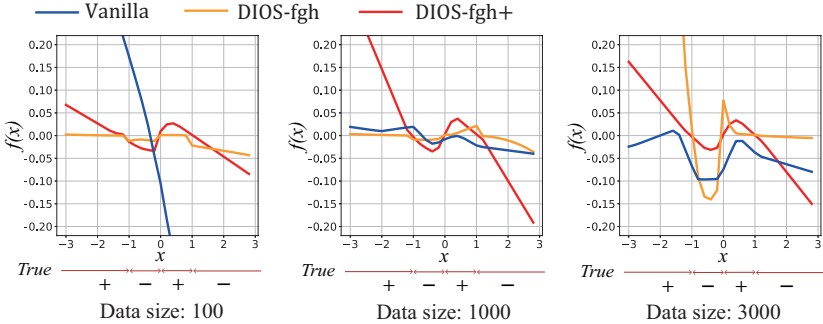

Figure 4: The sketch $x$-$f(x)$ displaying the internal dynamics trained from the bistable datasets with the different sizes. The sign of the bistable model is shown at the bottom of each figure.

the smaller RMSE for all methods. The vanilla neural networks do not consider the $\mathcal{L}_2$ gain, so the GainIO error was large in the case of the low RMSE.

Figure 4 shows the relationship between $x$ and $f(x)$ in the trained system in (1) to compare the vanilla neural networks, DIOS-fgh and DIOS-fgh+. DIOS-fgh and DIOS-fgh+ successfully find two stable points, i.e., the trained function $f(x)$ had roots of two stable points $x = \pm 1$ and an unstable point $x = 0$. Especially, these stable points were robustly estimated in the trained system using our presented loss function (DIOS-fgh+) even when the dataset size was small. The vanilla neural networks failed to obtain the two stable points in all cases.

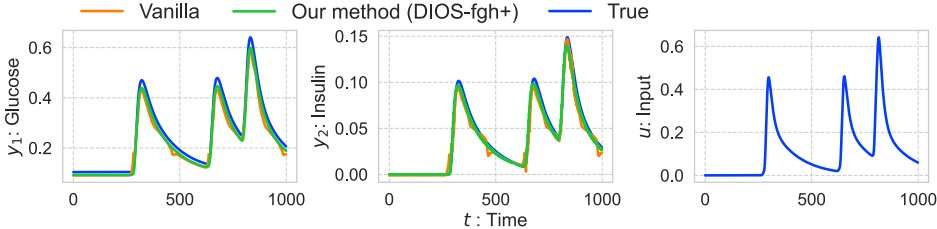

Figure 5: The input and output signals of the glucose-insulin simulator and the predicted output.

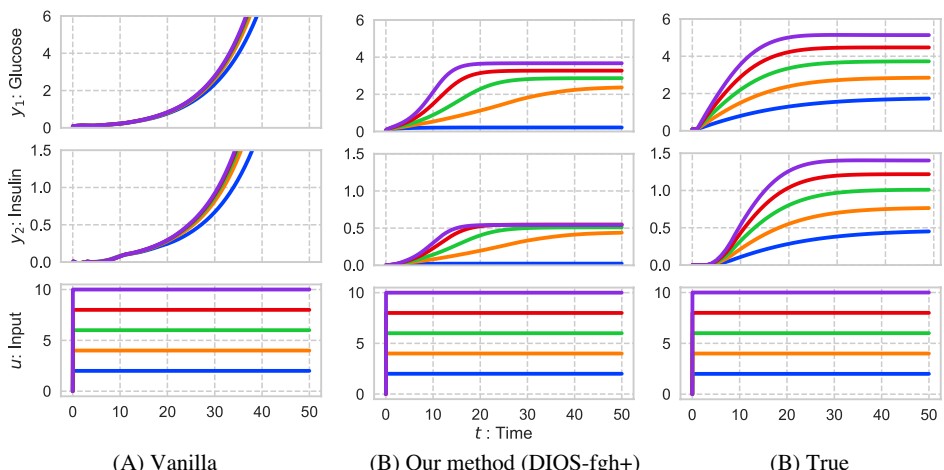

(A) Vanilla         (B) Our method (DIOS-fgh+)         (B) True

Figure 6: The step reaction of the trained systems. The color of each line indicates the magnitude of the step input.

## 5.3 Glucose-insulin benchmark

This section addresses an example of the identification of biological systems. We consider the task of learning the glucose-insulin system using a simulator [32] to construct responses for various inputs and evaluate the robustness of the proposed method for unexpected inputs. This simulator outputs the concentrations of plasma glucose $y_1$ and insulin $y_2$ for the appearance of plasma glucose per minute $u$. To determine the realistic input $u$, we adopt another model, an oral glucose absorption model [33].

Using this simulator, 1000 input and output signals are synthesized for this experiment. The input and output signals are sampled with a sampling interval $\Delta t = 1$ and 1000 steps for each sequence. In this benchmark, we set the number of dimensions of the internal system as six, fix a positive definite function $V(x) = x^2$, and use our loss function with $\lambda = 0.001$ and $\alpha = 0.001$. Training one model for this examples (1000 input signal) tooks about 7.5hours using 1GPU. The hyperparameters including the number of layers in the neural networks, the learning rate, optimizer, and the weighted decay are determined using the tree-structured Parzen estimator (TPE) implemented in Optuna [34].

The RMSE of vanilla and our proposed method are 0.0103 and 0.0050, respectively. So, from the perspective of RMSE, these methods achieved almost the same performance. Figure 5 shows input and output signals in the test dataset and the predicted output by vanilla neural networks and our method (DIOS-fgh+). The $\mathcal{L}_2$ stability of the system using the vanilla neural networks is not guaranteed. Since the proposed method guarantees the $\mathcal{L}_2$ stability, the output signals of DIOS-fgh+ are bounded even if the input signals are unexpectedly large.

To demonstrate this, we conducted an additional experiment using the trained system. Figure 6 shows the transition of output behavior caused by the magnitude of the input signal changing from 2 to 10. Note that the maximum magnitude in the training dataset is one. In this experiment, $\Delta t$ was changed to 0.01 and the clipping operation was removed to deal with the large values of the state.

This result shows the output of the vanilla neural networks quickly diverged with an unexpectedly large input. Contrastingly, the output behavior of our proposed method is always bounded. Therefore, we actually confirmed that our proposed method satisfies the $\mathcal{L}_2$ stability.

## 6  Conclusion

This paper proposed a learning method for nonlinear dynamical system guaranteeing the $\mathcal{L}_2$ stability. By theoretically deriving the projection of a triplet $(f, G, h)$ to the space satisfying the Hamilton-Jacobi inequality, our proposed method realized the $\mathcal{L}_2$ stability of trained systems. Also, we introduced a loss function to empirically achieve a smaller $\mathcal{L}_2$ gain while reducing prediction errors. We conducted two experiments to learn dynamical systems consisting of neural networks. The first experiment used a nonlinear model with multiple asymptotically equilibrium points. The result of this experiment showed that our proposed method can robustly estimate a system with multiple stable points. In the next experiment, we applied our method to a biological system using a simulator of the glucose-insulin system. It was confirmed that the proposed method can successfully learn a proper system that works well under unexpectedly large inputs due to the $\mathcal{L}_2$ stability. There is a limitation that our method cannot apply the system without the $\mathcal{L}_2$ stability. Future work will expand the $\mathcal{L}_2$ stability-based method to a more generalized learning method by dissipativity and apply our approach of this study to stochastic systems.

## Acknowledgements

This research was supported by JST Moonshot R&D Grant Number JPMJMS2021 and JPMJMS2024. This work was also supported by JSPS KAKENHI Grant No.21H04905 and CREST Grant Number JPMJCR22D3, Japan. This paper was also based on a part of results obtained from a project commissioned by the New Energy and Industrial Technology Development Organization (NEDO).

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
