# OpenReview forum: " Learning Deep Input-Output Stable Dynamics"
_NeurIPS.cc/2022/Conference — NeurIPS 2022 Accept_

### Official Review · Reviewer_p8Kb · 2022-06-26

**Rating:** 5
**Confidence:** 4
**Ethics Flag:** Yes
**Soundness:** 2 fair
**Presentation:** 3 good
**Contribution:** 2 fair

**Summary:**

Noticing the input-output stability in dynamical systems, the paper proposed a projection method to enforce the Hamilton-Jacobi inequality in learning such dynamics with neural network. The projection problem was converted into a quadratic constraint quadratic programming problem and solved analytically. The paper also provided experimental results to show the effectiveness of such methods.

**Questions:**

1. I’m not convinced by the introductory examples in figure 1. The author claims the reason is “magnitude of this input signal was larger than that of the input signals in the training dataset”. However, it looks like the output signal is unbounded (thus not satisfying I/O stability under finite step activation) regardless of input signal magnitude.
2. As for notation, the author denotes $(\hat{f},\hat{G},\hat{h})$ as nominal dynamics and $(f,G,h)$ as the modified dynamics in 3.1. It is confusing since $(f,G,h)$ denotes the true dynamics in equation 1. I suggest using other styles.
3. In the loss function, why the 2nd term is needed here? Since $\hat{y}$ is calculated by modified $(f,G,h)$, then such trajectory must satisfy the Hamilton Jacobian inequality. nominal $(\hat{f},\hat{G},\hat{h})$ is not directly used in prediction, why do we need to penalize nominal $(\hat{f},\hat{G},\hat{h})$?
4. Regarding the computational time, I’m curious what is bottleneck for such long hours (60 GPU day in total). The toy example is simply single variable ODE, the Glucose-insulin examples only have 1000 input and output signals and seems to be low dimensional as well.

**Limitations:**

There is no potential negative societal impact for this work.

**Strengths And Weaknesses:**

Strengths:
1. The paper structure is easy to follow and the hierarchy of math derivation is clear. The experiments are well organized and helpful for theorem illustration.
2. Despite many recent papers to enforce certain properties in learning dynamics, the input-output stability idea is novel.


Weakness

1. The problem formulation (eqn 1) with $ \dot{x} $ linearly dependent on u limits the application to certain systems. But this is common treatment in dynamical system literature and is not a big concern.

2. The projection idea can be seen as a continuum of [A],[B], with modification on projection methods to enforce Lyapunov like properties. The setups in the above two papers are more flexible, but lack the input u and h() output function. The projection method in this paper essentially downgrades to the setup in [A] with y=x and u=0. I’m not confident but the similar model performance might also be achieved from previous works. Please correct me if I'm wrong or provide comparison experiments.

[A]: Gaurav Manek and J. Zico Kolter. Learning stable deep dynamics models. In Advances in Neural 298 Information Processing Systems (NeurIPS), volume 32, 2019.
[B]Nathan Lawrence, Philip Loewen, Michael Forbes, Johan Backstrom, and Bhushan Gopaluni. Almost 300 surely stable deep dynamics. In Advances in Neural Information Processing Systems (NeurIPS), volume 33, 301 2020.

3. In the main experimental results(figure 1 and 6), I’m concerned if the bad performance of the step input response is mainly due to model never been unexposed to such input during the training process. The results' improvement from the proposed method could be trivial if adding similar examples into training data could address such issue.

4. the experiments could be stronger to illustrate the application of the method on more complex and diverse examples. The toy example is too simple, and the general audience are not familiar with the latter glucose-insulin system in terms of complexity, modeling difficulty, system stability etc. The current examples are not representative. This is my major concern.

5. More experimental details are needed, such as neural network structure, total parameter number in both NN and other systemID methods, training hyperparameters etc.

---

> ### Author Response · Authors · 2022-08-02
> **Response to Reviewer Comments**
>
> Thank you for the preliminary review. We’d like to use this opportunity to correct some inaccuracies or misconceptions in it.
>
> > Weakness1 :
>
> That is correct. We used a commonly used input-linear model.
>
> > Weakness2 :
>
> We are concerned that the reviewer may have misunderstood the relationship between input-output stability in this paper and internal stability in [A][B].
> As you say, (a) to make the input-output system consistent with [A], we need to assume y=x,u=0.
> (b) Even if internal stability is guaranteed by [A] and other methods, input-output stability cannot be achieved.
>
> (a) We can say that Theorem 1 is a generalization in reference [A]. Therefore, when k=0, we obtain the same equation as in [A]. Strictly speaking, the reference [A] shows the case of \nable V f \leq \alpha V, whereas we show the case of \nable V f \leq 0. Our method can be simply extended to this different version.
>
> (b) More strictly, if the input-output system is observable, input-output stability guarantees internal stability. Our method therefore theoretically guarantees a more strict condition.
> If we were to simply fit the existing methods to the observed data, the system would have zero inputs, which is a trivial system ignoring the effects of the inputs; therefore, it is inconsistent with our target input-output systems.
> Also, if we apply the projection of the existing method to our target system, we can just guarantee an internal stable system, but not an input-output stable system.
>
> > Weakness3 :
>
> We agree with the first part. We chose an extreme example, where different types of inputs for training and testing were used, for clarity. Our method is not limited to such cases and can guarantee input-output stability for any unknown input.
> In actual applications, it is not realistic to prepare data assuming every input in advance. So, we believe that methods for guaranteeing some properties against even unknown inputs like our method are valuable.
>
>
> > Weakness4 :
>
> We agree that the method has potential applicability to a wider range of applications.
> First, we emphasize that our toy model contains the following three important insights:
> - Classical linear systems cannot learn multiple stabilities like bistable dynamics.
> - The previous study [A] only focuses on a unique stable point, so this toy model cannot be applied.
> - It suggests that the vanilla method cannot learn input-output stable dynamics, even for too simple examples like this.
>
> In the next experiments, we chose a biological system as a more realistic example, which is a typical and important input-output (stable) system as described in Introduction, for readers interested in the input-output stable systems. In response to your suggestion, we have also added an explanation of this biological model in Appendix E to help the general reader understand the glucose-insulin system.
>
>
> > Weakness5 :
>
> Thank you for your advice.
> We added the details related to the hyperparameters in Appendix D.
>
> > Q1.
>
> We intended to represent the L2 norm (integral value) of the time series signal as “magnitude”.
> As you pointed out, we clarified that because it was confusing.
>
> > Q2.
>
> Thank you for your valuable advice. We modified the notation to clarify this point:
> (f_n,G_n,h_n) : nominal dynamics
> (f_m,G_m,h_m) : modified dynamics
>
> > Q3
>
> This term is used to stabilize the computation in the training process.
> Since the prediction error is calculated only by the modified dynamics (f, G, h), the nominal dynamics (f^, G^, h^) have degrees of freedom.
> By using this term, we expect that learning will be stable by eliminating the degree of freedom of (f^, G^, h^).
>
>
> > Q4.
>
> This is because the total execution time is measured in multiple experiments.
>
> For the one experiment for the toy example with 1000 input signals, it took about 1 hour using 1 GPU to learn one model. Training a glucose-insulin experiment with 1000 input signals took 7.5hours using 1GPU.
> Also, with 3000 input signals, each experiment takes about three times as long.
> The computational bottleneck is empirically the computational cost depending on the sequence length (or Δt) in our straightforward implementations.

---

> > ### Comment · Reviewer_p8Kb · 2022-08-07
> > **response to author rebuttal**
> >
> > Thank you for the detailed reply and corresponding revision. That clarified most of my concerns.
> >
> > Regarding weakness 3, I agree it is valuable for a method to guarantee certain properties on unknown data since not all of them are observable during training. For this task with potentially unstable behavior, I think at least the model should be exposed with certain amount of unstable examples. It is understandable that authors introduce this toy example as an extreme case. However, in the real world scenario, such trajectories could also be observed and the performance issue might be addressed by allowing a few unstable cases in the training data.
> >
> > Regarding the biological system, thank you for providing the detail experiment description. I still suggest some more intermediate examples for general audience, such as the environments provided by OpenAI's gym package or other common dynamics examples.
> >
> > I will increase my score while leaving the decision to reviewer discussions.

---

> > > ### Author Response · Authors · 2022-08-08
> > > **We appreciate the thoughtful response**
> > >
> > > We thank the reviewer for their valuable time and insightful comments.
> > > We think the suggested changes and additions made here have improved the work, and they will be included in the final version. Because this software will be released as open source, we would consider using a de-facto standard package and providing some more intermediate examples for general audiences. In addition to a step function, which often examines the input-output gain, we also consider adding some realistic input examples including white noise and square waves.

---

### Official Review · Reviewer_adB5 · 2022-07-10

**Rating:** 7
**Confidence:** 5
**Soundness:** 2 fair
**Presentation:** 3 good
**Contribution:** 3 good

**Summary:**

They propose an approach that uses the Hamiltonian-Jacobi (HJ) inequality to guarantee that the learned nonlinear dynamics have input-output stability.

This problem is fundamental and important in the sense that it bridges the gap between theoretical control theory and data-driven control for unknown nonlinear systems using well-established tools from control theory like the HJ inequality.


**Questions:**

1) What are the assumptions for (f,G,h) after Eqn(1)? The HJ inequality does not guarantee input-output stability if (f, G,h) are not continuous or differentiable. Please recheck.

2) Please cite the relevant reference for Proposition 1

3) As I am aware of the HJ inequality, apart from V being a positive definite function (which is mentioned), V must also be continuously differentiable and the value of V(0) must be zero which I think is not mentioned in your manuscript.

4) Guaranteeing that the HJ inequality is satisfied over dataset D (by minimizing the loss function given in (9)) does not ensure that the HJ inequality is satisfied for all points in the feasible domain of (x0,u,y). This is a major challenge that I observed in many recent relevant papers. However, some papers effectively address this issue by using SMT solvers where the SMT solver generates counter-examples that do not satisfy the inequality and are therefore fed back to the neural network to train. Please see relevant refs titled “Neural Lyapunov Control”, “Neural Koopman Lyapunov Control”, and “Neural Lyapunov Redesign” and the relevant references therein.

5) In eqn after (9), what does R(HJ(..)) denote? It is not clear to me from the write-up.

6) Please give some insight on whether one can compute a valid V that satisfies the HJ inequality. I think that is not mentioned in the manuscript except in the part where you write “The function V can be designed as a trainable neural network [14] or as a fixed function that achieves the desired properties of the nonlinear dynamics [15]”. However, this is not very clear to me.

7) Please give a brief remark on how you compute yhat from the nominal dynamics (f, G,h) given (x0,u,y) in the simulations.
I am wondering why is “Relevant work” in Section 4 and not after Section 1?


**Limitations:**

The main limitation of the approach is point 4 under “Questions” and it is also not clear to me on know do they compute a valid V.


**Strengths And Weaknesses:**

The merits of the paper are as follows:
1) In contrast to the most recent approaches which simultaneously design a controller and learn the dynamics, this paper goes a step ahead to also ensure input-output stability for the learned dynamics using the well-established HJ inequality.

2) The results are extensive in the sense that they clearly validate their approach and compare it with other state-of-the-art and recent approaches.

The weakness of the paper is as follows:
1) Ensuring that the HJ inequality is satisfied over a finite dataset D does not guarantee the same is true over the entire feasible time
Assumptions of (f, G, h) are not clear

More questions are in the next section “Questions”

---

> ### Author Response · Authors · 2022-08-02
> **Response to Reviewer Comments**
>
> We appreciate the reviewer for their time and insightful comments. We will improve our paper based on the advice and comments of the reviewer.
>
> > Q1
>
> The Hamiltonian-Jacobi inequality needs continuous differentiability of V, locally Lipschitz of f, and continuity of (G, h). These conditions are satisfied because the nominal dynamics (hat f, hat G, hat h)  are represented by NNs and the modification of our method (Theorem 1, Corollary 1,2) is continuous.  For clarification, these conditions are added in Proposition1.
>
> > Q2
>
> Thank you for your advice. I added it.
>
> > Q3
>
> The description of V was written as “V is a smooth positive definite function” on line 77 in Background. We would like to rewrite it more strictly as “continuously differentiable”. V(0) = 0 was described on line 74.
>
> > Q4
>
> The first sentence of your question is correct, but we are concerned that the reviewer may have misunderstood our claim. We do not claim that simply minimizing the loss function means satisfying the Hamilton-Jacobi inequality at all points. We claim that our method analytically guarantees the Hamilton-Jacobi inequality at all points by our modification method (Theorem 1, Corollary 1,2) by projecting the nominal dynamics (hat f, hat G, hat h). This loss function (and the second term in (9)) aids the modification method by promoting the learning process.
>
> Thank you also for the fascinating related methods. We would like to add  the related work and the following discussion:
>
> ===
>
> Lyapunov functions are also used to design controllers, where the whole system should satisfy the stability condition. A method for learning such a controller using neural nets has been proposed [Neural Lyapunov Control]. This method deals with optimization problems over the space that satisfies the stability condition, which is similar to our method. Whereas we solve the QCQP problem to derive the projection onto the space, this method uses an SMT solver to satisfy this condition. The method has also been extended to apply unknown systems [Neural Koopman Lyapunov Control].
>
> ===
>
> “Neural Lyapunov Redesign” is an important method for rigorously estimating the region of attraction, but we think the discussion of the region of attraction is beyond the scope of our work.
>
>
> > Q5
>
> When the nominal dynamics do not satisfy the HJ equations, this term becomes a positive value. To realize this, we assign the nominal dynamics (\hat{f},\hat{G},\hat{h}) to the argument of HJ, where HJ is defined at line 101 and  R is the ramp function, defined at line 100. In the learning phase, this term, therefore, has the effect of bringing the nominal dynamics closer to the modified dynamics.
>
> > Q6
>
> We would like to add an explanation of how to determine V to clarify this point.
> Since this method learns dynamics that satisfy the Hamilton-Jacobi inequality with respect to a given V, we can design V depending on the desired property of the target dynamics.
> We added the concrete example related to a recommended way to design V at line 106.
>
> ===
>
> A way to design V is to determine the desired positive invariant set and design the increasing function around this set. For example, if the target system has a unique stable point, x=0 can be regarded as the positive invariant set and the increasing function V can be designed as V(x) = 1/2x^2.
>
> ===
>
>
> There is also a method of learning V from data, as in [14], but since it would be misleading as you point out, we put this discussion in Related work:
>
> ===
>
> In this paper, the Lyapunov function V is considered to be given, but this function can be learned from data. Since a pair of dynamics (1) and V has redundant degrees of freedom, additional assumptions are required to determine V uniquely. In [9], it is realized by limiting the dynamics to the internal system and restricting V to a convex function [14].
>
> ===
>
>
> > Q7
>
> yhat is derived by the Euler method using the initial value x0 and the input u according to the “modified" dynamics (f,G,h).  Specifically, I added this procedure to Algorithm 1,2 in Appendix~C.
>
> Since this paper spans several fields including dynamical systems and machine learning, the volume of “related work” is large to put before “Method”.

---

> > ### Comment · Reviewer_adB5 · 2022-08-08
> > **Response to authors**
> >
> > All my comments have been addressed. I would be increasing my score significantly.

---

> > > ### Author Response · Authors · 2022-08-09
> > > **Thanks for your response**
> > >
> > > We appreciate your additional efforts and time to carefully read our response. We are happy to hear that our response successfully addressed your concerns.

---

### Official Review · Reviewer_shMz · 2022-07-12

**Rating:** 5
**Confidence:** 4
**Soundness:** 3 good
**Presentation:** 2 fair
**Contribution:** 3 good

**Summary:**

This is a system identification method for nonlinear dynamical systems represented as ODEs based on input and output signals. Authors use neural network for such a system identification. The novelty of the paper is that the identification also comes with the guarantee on the stability. The enabling technique is to enforce/projection onto Hamiltonian-Jacobi inequality manifold for L-2 stability.

**Questions:**

Can you clarify on the application of this method to chaotic ODEs? Is it a limitation? What challenges do you foresee?

Is it fair to say, this work is an extension of Neural ODE to enforce stability? If not, can authors briefly comment on such a possibility? Seems like this method, since it uses Euler method, can be faster than Neural ODE. What about generalization? Also what are the impact of $\Delta T$ on the convergence and numerical stability of learning.

It's a bit confusing that \hat{y} is generated by modified dynamics (f,G,h); while the nominal dynamics is denoted by (\hat{f},\hat{G},\hat{h}). Is there a reason authors used such notation?

Maybe a schematic of the method or an algorithm can help to better clarify the method. It is still unclear how do authors in general prefer to represent V.

Is it possible that the original system itself is not stable and then this system stabilize the neural network representation of that?

**Limitations:**

Yes.

**Strengths And Weaknesses:**

Strengths:
System identification with neural networks has the potential to better generalize than existing methods. However, there is no guarantee that such methods are stable. This paper brings ideas from control and dynamical systems community and apply them to the learning-based system identification to enforce L2 stability. The resulting networks can be used for prediction of dynamical systems in the long-term.

Weaknesses:
The method can be applied to nonlinear dynamical systems with fixed points or limit cycles. It is not clear if the method is applicable to more generic dynamical systems with strange attractors.
The method described is still a bit unclear.
Although authors compare with several DNN architectures, more comparison should be made with traditional system identification methods rather than deep-learning based methods (e.g. ERA, SINDY, DMD, etc.).

---

> ### Author Response · Authors · 2022-08-02
> **Response to Reviewer Comments**
>
> Thanks for your valuable comments.
> We have split your comments into the following questions and answer them one by one. We hope the following responses can address your concerns.
>
> > Weaknesses:
>
> Thank you for introducing a beneficial method. We believe that ERA, which identifies a linear state-space model for impulse inputs, is important from a viewpoint of traditional system identification. DMDc, a linear system identification method, and its nonlinear extension SINDYc focus on non-state-space models, which is not the target of this paper. Because these presented methods are commonly used in the field of dynamical systems, we added the description of them to line 158 in Related Work:
>
> ===
>
> Linear state-space models include identification methods by impulse response like the Eigensystem Realization Algorithm (ERA), by the state-space representation like Multivariable Output Error State sPace (MOESP) and ORThogonal decomposition method (ORT). System identification methods for non-state-space models, unlike our target system, contain Dynamic Mode Decomposition with control (DMDc) and its nonlinear version, Sparse Identification of Nonlinear DYnamics with control (SINDYc).
>
> ===
>
> > Q1 Can you clarify on the application of this method to chaotic ODEs? Is it a limitation? What challenges do you foresee?
>
> We believe that it is theoretically possible to apply this method to positively invariant sets containing strange attractors since strange attractors are positively invariant sets. For example, S\subset \R^n is a candidate of strange attractor, then V:= inf_{y in S}\|y - x\|^2 In this case, the positive definite function V is flat over S in the state-space. The state-space dynamical system (1) has degree of freedom and can be rescaled in the region of state space corresponding to the output. Thus, our method cannot constrain dynamics (f,G,h) over the flat S. A possible future work is a derivation of a new modification theorem to restrict the  degree of freedom in such cases.
>
> Also, there is an optimization challenge in "learning" complicated models such as chaotic ODEs from finite data while maintaining input-output stability with NNs.
>
>
> > Q2 Is it fair to say, this work is an extension of Neural ODE to enforce stability? If not, can authors briefly comment on such a possibility? Seems like this method, since it uses Euler method, can be faster than Neural ODE. What about generalization? Also what are the impact of ΔT on the convergence and numerical stability of learning.
>
> We can say that our work is an extension of Neural ODEs with the Euler method to enforce input-output stability. As you say, many more advanced methods have been proposed in the context of Neural ODEs.This method is not limited to the Euler method, but can be applied to other methods such as Runge-Kutta methods, and can be used together with the various techniques for the Neural ODEs. To exclude convergence issues from our discussion, we chose a sufficiently small value for ΔT as far as computational resources allow. It is necessary to select a smaller ΔT for the stability of numerical calculations in the early stage of learning rather than in the model after training. In practical applications, it is expected that the choice of ΔT (and the choice of discretization algorithm) will be important considering the trade-off with computational speed, as is well known in numerical computations.
>
> > Q3 It's a bit confusing that \hat{y} is generated by modified dynamics (f,G,h); while the nominal dynamics is denoted by (\hat{f},\hat{G},\hat{h}). Is there a reason authors used such notation?
>
> Thank you for your advice. Although we follow the notation of the previous paper, we denoted the nominal dynamics as (f_n, G_n, h_n) and the modified dynamics as (f_m, G_m, h_m) for easier understanding in our context.
>
> > Q4 Maybe a schematic of the method or an algorithm can help to better clarify the method. It is still unclear how do authors in general prefer to represent V.
>
> Thank you for your suggestion.
> We added the algorithm scheme to Appendix C.
> Although our method does not have limitations on the design of V,  it is preferable to design a positive invariant set and a function that increases around it as a practically meaningful selection method of V. In our experiment, we designed V in this way.
> We added the description about the design of V on line 103.
>
> > Q5 Is it possible that the original system itself is not stable and then this system stabilize the neural network representation of that?
>
> Yes. Our method selects a model by minimizing the prediction error in the space (f, G, h) where input-output stability is satisfied. If the original system is not input-output stable, the error is expected to grow.

---

### Review · Ethics_Reviewer_rmPz · 2022-08-05

**Recommendation:**

Though the work does not raise ethical issues to me, a bit more could be said to satisfy comments on the impact of these methods in society.  Given that I believe this is more foundational work, I would suggest the authors briefly acknowledge some of the potential broader impacts in either their introduction or conclusion. See https://neurips.cc/public/guides/PaperChecklist section (c).

**Ethics Review:**

While the reviewers concerns may be valid with respect to limited evaluation of the methods, I'm not sure that there are broader ethical concerns as outlined by the Ethics Guidelines.

In terms of ethical evaluations, it does not appear that the authors discussed the potential impacts of the work as requested in (c) of the checklist that asks:

"We expect many papers to be foundational research and not tied to particular applications, let alone deployments, but being foundational does not imply that research has no societal impacts. If you see a direct path to any negative applications, you should point it out, even if it's not specific to your work. In a theoretical paper on algorithmic fairness, you might caution against overreliance on mathematical metrics for quantifying fairness and examples of ways this can go wrong. If you improve the quality of generative models, you might point out that your approach can be used to generate Deepfakes for disinformation. On the other hand, if you develop a generic algorithm for optimizing neural networks, you do not need to mention that this could enable people to train models that generate Deepfakes faster."

The authors discuss how the methods connect to real world applications, but not the potential for societal impacts.  No negative impacts immediately come to mind based on my reading, but I do note it does not appear discussed.

---

### Review · Ethics_Reviewer_zZeY · 2022-08-07

**Recommendation:**

the authors could include an acknowledgment of potential negative impacts in the current version.

**Ethics Review:**

the paper's evaluation is in line with its claims and it does not use sensitive data

---

### Meta-Review · Program_Chairs · 2022-09-13

**Recommendation:** Accept
**Confidence:** Certain

**Metareview:**

This paper proposes a method to learn nonlinear dynamic systems with certain theoretical guarantees.

The reviews express a positive evaluation for the research question and that the methods proposed generalise better than current SOTA. There are some concerns as to whether some of the guarantees proposed were truly verifiable, but these were resolved in the discussion phase.

The ethics review makes recommendations around including potential negative societal impacts, which we hope the authors would take on board.

**Award:**

No

---

### Decision · Program_Chairs · 2022-09-14

Accept